# Exploring the Early Stages of the Amyloid Aβ(1–42) Peptide Aggregation Process: An NMR Study

**DOI:** 10.3390/ph14080732

**Published:** 2021-07-27

**Authors:** Angelo Santoro, Manuela Grimaldi, Michela Buonocore, Ilaria Stillitano, Anna Maria D’Ursi

**Affiliations:** Department of Pharmacy, University of Salerno, Via Giovanni Paolo II, 132-84084 Fisciano, SA, Italy; asantoro@unisa.it (A.S.); magrimaldi@unisa.it (M.G.); mbuonocore@unisa.it (M.B.); istillitano@unisa.it (I.S.)

**Keywords:** Alzheimer’s disease, amyloid peptide, NMR structure, Aβ(1–42), molecular dynamics

## Abstract

Alzheimer’s disease (AD) is a neurodegenerative pathology characterized by the presence of neurofibrillary tangles and amyloid plaques, the latter mainly composed of Aβ(1–40) and Aβ(1–42) peptides. The control of the Aβ aggregation process as a therapeutic strategy for AD has prompted the interest to investigate the conformation of the Aβ peptides, taking advantage of computational and experimental techniques. Mixtures composed of systematically different proportions of HFIP and water have been used to monitor, by NMR, the conformational transition of the Aβ(1–42) from soluble α-helical structure to β-sheet aggregates. In the previous studies, 50/50 HFIP/water proportion emerged as the solution condition where the first evident Aβ(1–42) conformational changes occur. In the hypothesis that this solvent reproduces the best condition to catch transitional helical-β-sheet Aβ(1–42) conformations, in this study, we report an extensive NMR conformational analysis of Aβ(1–42) in 50/50 HFIP/water *v*/*v*. Aβ(1–42) structure was solved by us, giving evidence that the evolution of Aβ(1–42) peptide from helical to the β-sheet may follow unexpected routes. Molecular dynamics simulations confirm that the structural model we calculated represents a starting condition for amyloid fibrils formation.

## 1. Introduction

Neurodegenerative diseases represent an increasingly common debilitating condition due to the current average lifespan. Among the neurodegenerative pathologies, Alzheimer’s disease (AD) is one of the most studied [1,2,3]. This disorder is characterized, at the histological level, by the presence of neurofibrillary tangles and amyloid plaques, mainly composed of 39–42 amino acid-long peptides known as amyloid-β peptides (Aβ). Aβ(1–40) and Aβ(1–42), like other amyloid peptides—transytiretin, IAPP [4,5,6,7,8]—derive from a noncorrect cleavage of the transmembrane amyloid precursor protein (APP). They are in a metastable equilibrium at the neuronal cell membrane interface, where slight changes in the chemical–physical conditions—e.g., metal ions, pH, temperature—induce conformational transitions to form β-sheet structures, evolving in starch-like amyloid fibrils [9,10,11]. Accumulation of synaptotoxic and neurotoxic Aβ(1–42) fibrils on neurons and glial cells cause activation of proinflammatory cascades, leading to mitochondrial dysfunction and increased oxidative stress, which induces impairment of intracellular signaling pathways, deregulation of calcium metabolism, apoptosis, and cell death [12]. 

The control of the Aβ aggregation process as a therapeutic strategy for AD has prompted the interest to investigate the conformation of the Aβ peptides, taking advantage of computational and experimental techniques [13]; NMR spectroscopy has been extensively used to study the structure of Aβ(1–42) and its shorter fragments in several environmental conditions, where soluble helical structures are favored. Detergent micelles have been used to reproduce the cell membrane interface [14,15,16,17,18,19]: in sodium dodecyl sulfate (SDS) and lithium dodecyl sulfate (LiDS) micelles, the short fragments Aβ(12–28), Aβ(25–35), and Aβ(1–28) form regular α-helices [17,18], while Aβ(1–40) is characterized by helical-kink-helical structure. This structural motif is conserved in Aβ(16–35) in mixed dodecyl phosphocholine (DPC)/SDS micelles, where the stability of the structure depends on the charge of the micelle surface [20]. NMR analysis in an aqueous system revealed that Aβ(1–40) forms a 3_10_ helix on ^13^H–D^23^ residues and is unstructured in the N- and C-terminal regions [14]. On the other hand, Aβ(1–42) monomer contains ^8^S–V^24^ and ^28^K–V^38^ α-helices and a β-turn on ^25^G–K^28^ residues [21,22]. 

Mixtures of hexafluoroisopropanol (HFIP) and water in different proportions have been used to study, by NMR, the conformational properties of Aβ(1–42) in solvent systems characterized by different polarities. In HFIP/water 80/20 *v/v,* Aβ(1–42) is arranged in two helical segments, ^8^S–G^25^ and ^28^K–V^38^, interrupted by a ^25^G–S^26^, β-turn [21]; by decreasing HFIP proportion, i.e., in HFIP/water 30/70 *v*/*v*, Aβ(1–42) assumes prevalent bends structures preserving a single α-helical segment on ^11^E–L^17^ sequence (Figure 7c) [23]. Circular dichroism (CD) experiments performed on Aβ(1–42) in mixtures composed of systematically different proportions of HFIP and water show that in HFIP/water containing more than 90% water, Aβ(1–42) conformation shifts to β-sheet. Moreover, at 50/50 HFIP/water proportion, the first evident Aβ(1–42) conformational changes are visible [21]. On this basis, and in the hypothesis that this solvent, characterized by intermediate physicochemical properties between those previously used, represents the best condition to catch transitional helical-β-sheet Aβ(1–42) conformations [24,25], we decided to perform an extensive NMR conformational analysis of Aβ(1–42) in 50/50 HFIP/water. Aβ(1–42) structure solved in the present study, compared to those previously solved in 80/20 and 30/70 HFIP/water *v*/*v*, indicates that the evolution of Aβ(1–42) peptide from helical to β-sheet may follow unexpected routes. Molecular dynamics (MD) simulations confirm that the structural model we calculated represents a starting condition for amyloid fibrils formation. 

## 2. Results

### 2.1. NMR Structure Determination of Aβ(1–42) Peptide

We expressed and purified a recombinant Aβ(1–42) peptide following the protocol of Walsh et al. [26]. To check the aggregation state of the Aβ(1–42) peptide before the Aβ(1–42) structural characterization, we recorded the Diffusion Ordered SpectroscopY (DOSY) experiment (data not shown) [27]. The molecular weight of Aβ(1–42) calculated from DOSY spectrum and using 1,4-dioxane as internal reference was compatible with Aβ(1–42) in the monomeric form. 1D ^1^H, 2D total correlation spectroscopy (TOCSY) and Nuclear Overhauser Effect SpectroscopY (NOESY) spectra of Aβ(1–42) in a mixture of HFIP/water 50/50 *v*/*v* were collected on a Bruker 600MHz at 298 K. Then, ^1^H chemical shift assignments were carried out by iteratively analyzing TOCSY and NOESY spectra using SPARKY software [28]. The 1D and 2D ^1^H spectra (Figure 1) showed 42 well-dispersed amide chemical shifts and uniform resonance line widths according to the characteristics of a structured peptide. In addition, the sequential chemical shift assignment was performed according to the Wüthrich procedure [29].

Figure 2a summarizes the regular sequential and medium-range NOE effects, N,N(*i*,*i*+2), α,N(*i*,*i*+2), α,N(*i*,*i*+3), and α,β(*i*,*i*+3) observed in 2D-NOESY spectrum. NOE patterns are consistent with regular secondary structures on the N-terminal and central region of Aβ(1-42), while high dynamic, flexible conformations are present on the C-terminal region of the peptide. The Ramachandran plot in Figure 2b shows that the peptide assumes a right-handed α-helix predominantly and, to a lesser extent, a β-sheet conformation.

NOE data were translated into interprotonic distances using the CALIBA routine of CYANA 2.1 software [30] and then used as restraints for the NMR structure calculations. Table 1 summarizes the statistics for the final NMR ensemble [31]. 

Figure 3 shows the bundle of the best 10 Aβ(1–42) NMR structures derived from simulated annealing calculation using CYANA software [30]. The structures, deposited in the Protein Data Bank with the accession code PDB ID: **6SZF**, are superimposed on the backbone heavy atoms of ^1^D–H^14^ N-terminal residues and ^15^Q–S^26^ central residues, showing a root mean square deviation (RMSD) of backbone atom position of 1.19 Å and 0.94 Å, respectively; on the contrary, as predicted from analysis of NOE data, the C-terminus is much more disordered and dynamic.

Figure 4 shows a ribbon representation of the most representative NMR structure of Aβ(1–42) acquired in HFIP/water 50/50 *v*/*v*. The *Define Secondary Structure of Proteins* (DSSP) plot calculated by GROMACS [32] confirms the prevalence of regular α-helix conformations on ^18^V–V^24^ residues. In addition, alternation of turn and bend conformations are observable at the N-terminal region. 

### 2.2. Molecular Dynamics 

The computer simulation of the Aβ(1–42) α-helix—β-sheet transition is challenging. The development of innovative techniques such as discrete molecular dynamics (DMD) from NMR chemical shifts—metadynamics [33] or enhanced sampling techniques (REMD) [34]—represents a significant improvement, allowing to build reliable models of conformational transition. 

To integrate the NMR experimental data and to evaluate the stability of the Aβ(1–42) conformation solved in HFIP/water 50/50 *v*/*v* when exposed to a complete aqueous system, we performed 50 ns molecular dynamics (MD) calculation starting from the 3D coordinates obtained from NMR spectra assignment. Although aware of the limitations of the traditional computational methods we used, the results are in agreement with those obtained with REMD protocols [33,34,35,36,37]. 

Calculations were run with GROMACS [32,38]. The best NMR structure based on the lowest value of the target function (CYANA 2.1 [30]) was positioned in a box filled with explicit water for 50 ns at 298 K. As shown by the RMSD plot, the system reaches equilibrium after 1 ns and is stable during the whole simulation (Figure 5) [39]. 

The snapshots of the 50 ns MD trajectory of Aβ(1–42) in water are shown in Figure 6, overlapped to the solvent-accessible surface area (*sasa*) plot. As reported, the ^18^V–V^24^ α-helical structure slides to the ^12^V–V^18^ portion during the first steps of the dynamics, experimenting unfolding after 40 ns. Remarkable is the β-sheet architecture formed at C-terminus ^27^N–I^41^, which turns out to be extended to three strands within the 18 to 28 ns time range. Confirming data previously published [40], this is a crucial step for amyloid seeding, which is known to originate in the neighborhood of the ^25^G–M^35^ residues. From 28.1 ns to the end of MD simulation, this strand gets confined in residues ^35^M–V^40^, indicating that the fibrillation is occurring, and the β-strand is transforming into a more complex intermolecular structure. Solvent-accessible surface data (GROMACS *sasa,*
Figure 6) support this interpretation and show that N-terminal and central regions are equally exposed to the solvent until 40 ns, when the central part has low solvent accessibility because it is shielded by the N-terminal segment. 

## 3. Discussion

Based on the amyloid-β (Aβ) cascade hypothesis, abnormal accumulation of the Aβ peptides into toxic extracellular plaques is critical for inducing neurodegeneration and dementia in AD patients [9,41,42,43]. The main components of the amyloid plaques are Aβ(1–40) and Aβ(1–42), soluble helical peptides that, in overcoming toxic environmental conditions, undergo a conformational transition to form β-sheet oligomers, protofibrils, fibrils, and plaques [42]. How this transition occurs is still questioned. Experimental data on this process are derived from NMR studies of the peptide before and after the α-helix-β-sheet conformational transition, i.e., in solution [21,23] and the solid state [44,45]. Little data are available on the prevalent Aβ(1–42) conformations occurring in the intermediate steps of helical-to-sheet transition. 

HFIP/water mixtures have been used previously to solve the solution structure of Aβ(1–42), taking advantage of the physical–chemical properties of the fluorinated solvents [46]. In particular, these mixtures represented an excellent system to monitor the conformational perturbations induced by increasing amounts of water in the stable helical Aβ(1–42) solution structure. Aβ(1–42) studied by NMR in 80/20 HFIP/water *v*/*v* mixture [21] presented regular helical-kink-helical structure (Figure 7a), while in a mixture containing higher water content—30/70 HFIP/water *v*/*v*—it assumed bends conformations with regular α-helix on ^11^E–L^17^ segment (Figure 7c). 

In the present work, we analyzed by NMR, Aβ(1–42) peptide in 50/50 HFIP/water *v*/*v*, a solvent mixture characterized by intermediate polarity compared to those previously used. NMR data show that the prevalent conformation of Aβ(1–42) in these conditions includes helical structure on ^18^V–V^24^ residues, turns on ^4^F–H^14^ and ^32^I–G^38^ segments, short random coils, and bend structures on the remaining portions of the sequence (Figure 7b). The conformation of Aβ(1–42) peptide in 50/50 HFIP/water *v*/*v* is moderately flexible and characterized by intermediate regularity, as compared to the prevalent α-helix calculated in HFIP/water 80/20 *v*/*v* [21] and the prevalent bend structure calculated in HFIP/water 30/70 *v*/*v* [23].

However, comparing the localization of the helical segments, it is evident that the transition to a more extended structure, moving from 80/20 to 30/70 HFIP/water *v*/*v* condition, is not a simple unfolding process of the helical structure. Yet, it occurs through sliding the single α-helical pieces to the different segments of the sequence. Interestingly, by looking at the Ramachandran plot, a clusterized set of points is observable at psi = 70 and phi = −90 between the helical and sheet region. These points, characterized by average G-factor ranging from −2.40 to −1.62, correspond to the backbone dihedral angles of the residues flanking the helical region, thus evidencing the metastable condition of this region and its tendency to change conformation, favoring a further sliding of the helix.

To investigate the evolution of the NMR calculated structures in an aqueous medium, we performed 50 ns MD simulations. Snapshots of Aβ(1–42) trajectory throughout MD simulation and solvent accessible surface plot indicate that the ^18^V–V^24^ α-helix slides to residues ^12^V–V^18^ within the first steps, to be unfolded in the last steps of the MD. As soon as the system stabilizes (Figure 5 and Figure 6), a β-sheet architecture is formed in ^25^G–M^35^ segment, then shifts in the ^27^N–I^41^ region. These moieties are involved in forming a complex intermolecular structure, as shown by the low solvent accessibility of the ^35^M–V^40^ strand, which could indicate the rearrangement in a fibril aggregate structure.

## 4. Materials and Methods

### 4.1. Aβ(1–42) Peptide Production 

Aβ(1–42) peptide was obtained after transformation of *E. coli* BL21(DE3)-pLysS cells with PetSac plasmid, provided by the research group of Walsh [26]. To optimize expression levels, were used *E. coli*, Ca^2+^-competent (BL21(DE3)-pLysS) cells obtained by thermal shock and allowed to grow in agar plates containing the LB culture medium with ampicillin (50 mg/L). The individual colonies were taken from a stock solution, stored at a temperature of −80 °C, and added to 50 mL of preinoculum, where 50 μL of ampicillin (50 mg/L) was previously added. The preparation was left in a thermostatic incubator, under constant stirring, at a temperature of 37 °C overnight. The following day, 5 mL of the culture was transferred to 500 mL of LB medium added to 500 μL of ampicillin (50 mg/L). When bacterial clones reached a final OD_600_ of 0.6 at 37 °C, 50 mg/L of isopropyl-thio-β-D-galactoside (IPTG) was added. The cells were collected after 4 h of induction and centrifuged to remove the fraction constituted by the supernatant. Subsequently, the resulting pellet was thawed and sonicated three times in a solution containing Tris/HCl pH 8.0 and EDTA, at a concentration of 10 mM and 1 mM respectively, for 5 min on ice (1/2 horn, 50% duty cycle). The sonicate was centrifuged for 10 min at 8000 rcf. The supernatant was removed, and the pellet was resuspended in a solution containing 8 M urea, Tris/HCl pH 8.0 10 mM, EDTA 1 mM, and sonicate as before. The resulting solution, containing Aβ(1–42) in the urea-soluble inclusion bodies, was diluted with a 10 mM Tris/HCl solution pH 8.0 and 1 mM EDTA. Subsequently, the solution was purified with a HiTrap™ DEAE FF column, at a flow rate of 1 mL/min, using an AKTA purification system. The protein was eluted with an elution buffer (8 M urea, Tris/HCl pH 8.0 10 mM, 1 mM EDTA, 1 M NaCl). The eluted fraction was subsequently dialyzed against Tris/HCl pH 8.0 10 mM and freeze dried. The purity of the peptide was verified using the SDS-PAGE electrophoretic technique, electrophoresis on polyacrylamide gel (PAGE) in the presence of SDS, using Coomassie blue staining. The SDS-PAGE electrophoresis of the previously sonicated cell pellet revealed that the majority of Aβ(1–42) was present in the band between 4 and 5 kDa. In order to support the peptide expression, the matrix-assisted laser desorption/ionization (MALDI) mass spectrometry technique was used. The sample was previously dried and placed in a MALDI sample support consisting of a solution matrix (4-hydroxy acid, α-cyanic cinnamic, 25 mM citric acid), which favors the crystallization of the compounds and allows an optimal analysis.

### 4.2. NMR Sample Preparation

Aβ(1–42) peptide has a very high tendency to form fibrils. To preserve the peptide in the monomeric form and avoid the formation of oligomeric and polymeric forms, before and during the experiments, Aβ(1–42) was subjected to a defibrillating treatment following the procedure described by Jao et al. [47]. The Aβ(1–42) peptide was dissolved in trifluoroacetic acid (TFA) until complete solubilization of the powder sample and left in TFA for 3 h. Subsequently, the TFA was removed under nitrogen flow and was diluted 10-fold with milliQ water and lyophilized. This procedure was adopted for NMR analysis immediately before dissolution in the appropriate solvent.

### 4.3. NMR Spectroscopy

#### 4.3.1. Spectra Acquisition

NMR spectra were recorded on a Bruker DRX-600 spectrometer. Aβ(1–42) peptide (500 µM) was dissolved in HFIP/water-D_2_O 50/50 *v*/*v*. The 1D ^1^H homonuclear spectrum was recorded in the Fourier mode, with quadrature detection. The 2D ^1^H homonuclear TOCSY and NOESY experiments were acquired in the phase-sensitive mode using quadrature detection in ω1 by time-proportional phase incrementation of the initial pulse [48,49,50]. The water signal was suppressed by excitation sculpting experiments [51]. Before Fourier transformation, the time domain data matrices were multiplied by shifted sin^2^ functions in both dimensions. A mixing time of 80 ms was used for the TOCSY experiments. NOESY experiments were run at 298 K with mixing times of 200 ms. 

#### 4.3.2. Assignment of NMR Resonances 

The assignment of chemical shifts was obtained by the usual approach described by Wuthrich [29] examining 2D TOCSY and NOESY spectra using SPARKY software [28]. Intramolecular distance restraints derived from Nuclear Overhauser Enhancements (NOEs) were obtained from the 2D NOESY spectrum recorded on a 600 MHz spectrometer. 

#### 4.3.3. Structure Calculation 

Peak volumes were translated into upper distance bounds with the CALIBA routine from the CYANA 2.1 software package [30]. After discarding redundant and duplicated constraints, the final list of constraints was used to generate an ensemble of 50 structures by the standard CYANA protocol of simulated annealing in torsion angle space (using 6000 steps). The entries that presented the lowest target function value (0.83–1.19) and small residual violations (maximum violation = 0.38 Å) were analyzed using the PyMOL program [52].

### 4.4. Molecular Dynamics 

Molecular dynamics (MD) simulations on Aβ(1–42) structure obtained from CYANA calculations were performed with GROMACS [32,38], by using Gromos96 53a6 force field [53]. The simulations were run for 50 ns at 298 K. The structure was immersed in explicit water using the SPC model [54]. The protein was solvated, and the system was neutralized by adding three Na^+^ ions. After these steps, the energy minimization of the system was performed, and then the system was equilibrated using NVT and NPT runs. The temperature and pressure of the system were kept constant at 298 K and 1.01325 bar using the Berendsen weak coupling method [54,55]. The linear constraint solver (LINCS) method was used to constrain bond lengths [56]. The leap-frog algorithm was used to integrate the motion equations with a time step of 2 fs. The results were used for a 50 ns MD simulation using particle mesh Ewald for long range electrostatics under NPT conditions [57]. Coordinates were saved every 50,000 steps. The trajectory file was fitted in the box and converted into PDB coordinates by using the *trjconv* tool of the GROMACS package. The structure was visualized with Maestro by Schrödinger [58]. Root-mean-square deviation (RMSD) of peptide backbone was calculated using the *rms* tool of GROMACS, and plotted using R (version 3.6.0) [59]. Ramachandran plot was calculated using the *rama* tool of GROMACS. Solvent accessible surface plot was calculated using the *sasa* tool of GROMACS. The solvent surface accessible area graph was calculated by dividing the Aβ(1–42) peptide into three sections: N-terminus (residues ^1^D–Q^15^), central region (residues ^16^K–G^29^), and C-terminus (residues ^30^A–A^42^). Define secondary structure of proteins (DSSP) plot was calculated using the *do_dssp* tool of GROMACS.

## 5. Conclusions

We have studied the conformation of Aβ(1–42) in 50/50 HFIP/water *v*/*v* solvent. This mixture represents an intermediate condition between the previously explored apolar 80/20 HFIP/water *v*/*v* and polar 30/70 HFIP/water *v*/*v*. The structure we solved provides a snapshot of the conformation occurring during Aβ(1–42) helical-to-sheet transition: the apolar condition’s Aβ(1–42) loses its regular helix, which slides mostly to the central region, which is known to be critical for amyloid seeding. The C-terminus shows high flexibility and dynamic properties, revealing a primary role in the transition to the β-sheet conformation. Indeed, MD simulations in water confirm that these flexible regions evolve to regular β-strand structures, giving rise to a complex architecture typical of the β-sheet fibril aggregates.

## Figures and Tables

**Figure 1 pharmaceuticals-14-00732-f001:**
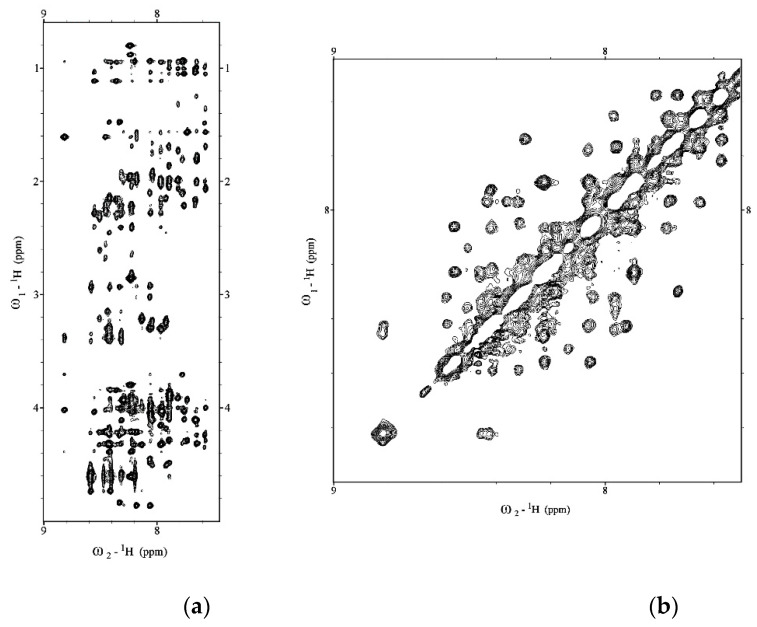
(**a**) Low field and (**b**) high field regions of 2D NOESY of Aβ(1–42) acquired on Bruker 600 MHz NMR in HFIP/water 50/50 *v*/*v*.

**Figure 2 pharmaceuticals-14-00732-f002:**
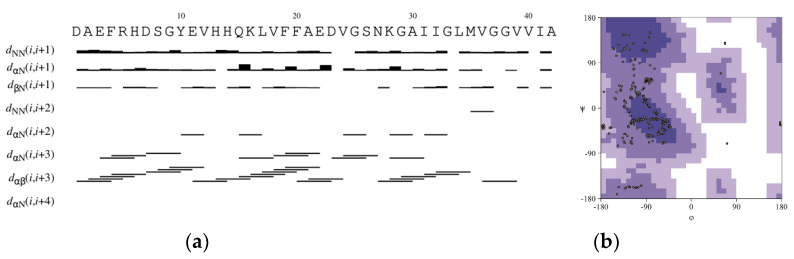
(**a**) Overview of the sequential and medium-range NOEs used to calculate the Aβ(1–42) peptide structure ensemble. (**b**) Ramachandran plot providing an overview of allowed and disallowed regions of torsion angle values.

**Figure 3 pharmaceuticals-14-00732-f003:**
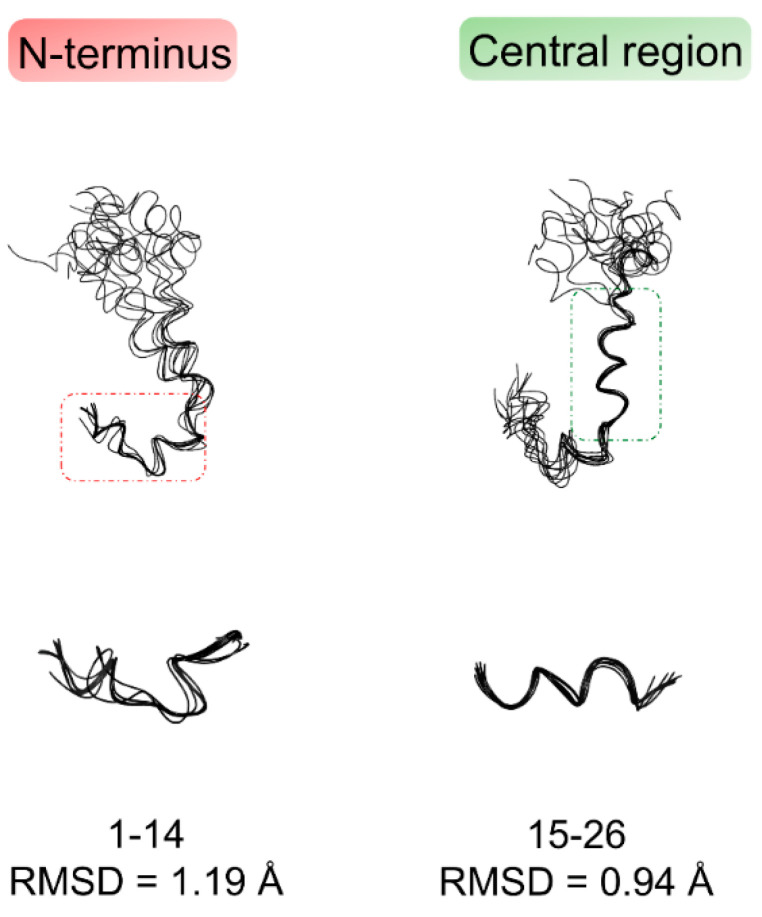
Bundles of the best 10 NMR structures of Aβ(1–42) derived from CYANA 2.1 [30] based on NMR data acquired in HFIP/water 50/50 *v*/*v*. The structures are superimposed on the backbone heavy atoms of N-terminal region ^1^D–H^14^ (left), and central region ^15^Q–S^26^ (right).

**Figure 4 pharmaceuticals-14-00732-f004:**
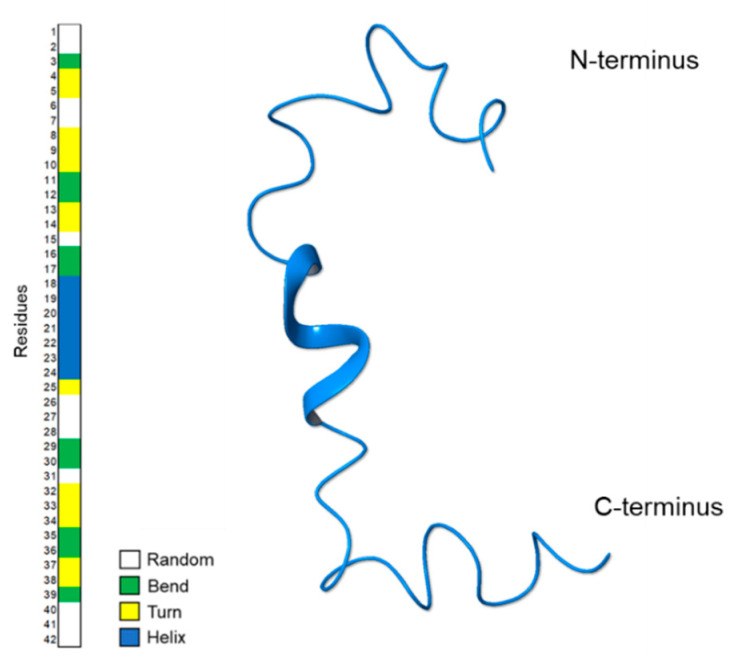
Ribbon representation of the most representative NMR structure of Aβ(1–42) acquired in HFIP/water 50/50 *v*/*v*. On the left, the *Define Secondary Structure of Proteins* (DSSP) plot calculated by GROMACS *do_dssp*.

**Figure 5 pharmaceuticals-14-00732-f005:**
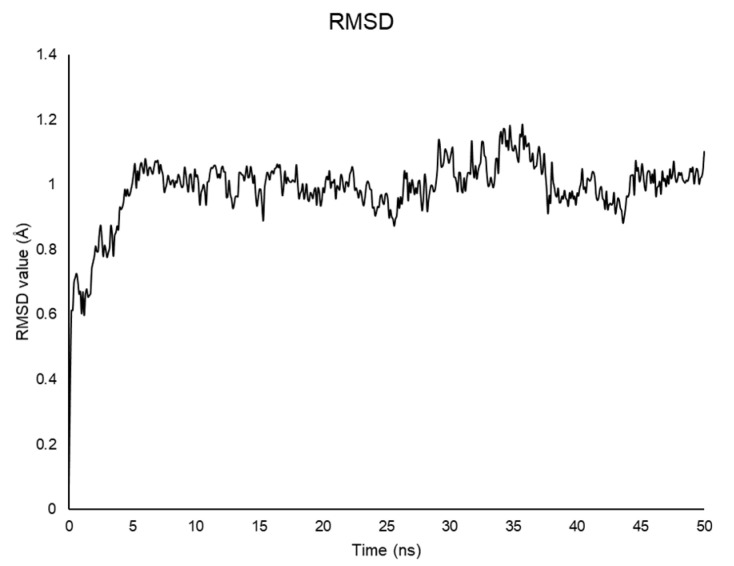
RMSD plot values for the backbone atoms of Aβ(1–42) throughout the 50 ns simulation as a function of time.

**Figure 6 pharmaceuticals-14-00732-f006:**
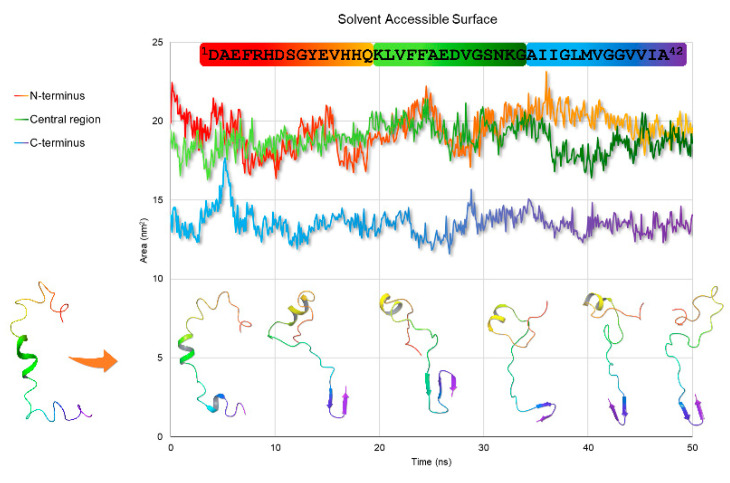
Snapshots of Aβ(1–42) trajectory throughout 50 ns MD simulation (on the bottom) superimposed to the solvent-accessible surface graph (on the top), according to the time step (ns) they were taken. The ribbons are colored in rainbow hue by atom position, from N-terminus (red) to C-terminus (purple).

**Figure 7 pharmaceuticals-14-00732-f007:**
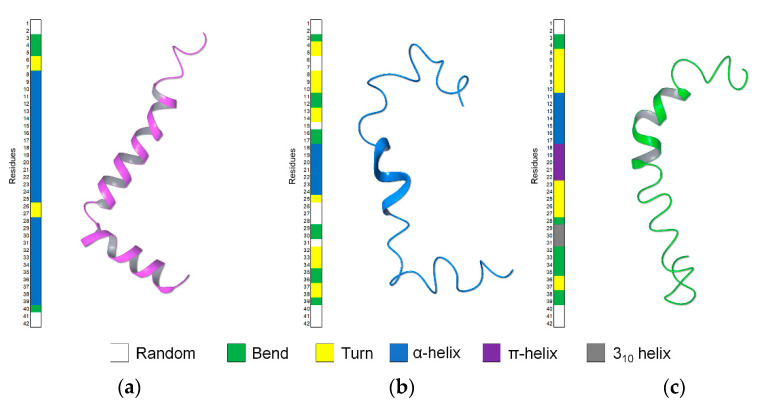
Comparison of the Aβ(1–42) structures in the different solvent system of HFIP/water *v*/*v*, (**a**) 80/20 (PDB ID: 1IYT); (**b**) 50/50 (PDB ID: 6SZF); (**c**) 30/70 (PDB ID: 1Z0Q).

**Table 1 pharmaceuticals-14-00732-t001:** Statistics for the structural calculation of the NMR ensemble of Aβ(1–42) peptide.

**Number of Experimental Restraints after CYANA**	
Total NOEs	585
Intra-residual	348
Sequential	143
Long-range	94
**RMSD**	
bb/heavy Å	2.92/3.11
**Ramachandran Analysis**	
Favorable regions	50.60%
Additional allowed regions	37.90%
Generously allowed regions	9.10%
Disallowed regions	2.4%

## Data Availability

PDB coordinates have been deposited in the Protein Data Bank. The access code is 6SZF. BMRB entry assigned accession number: 34440.

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
