# Peer review of "Exploring the Early Stages of the Amyloid Aβ(1–42) Peptide Aggregation Process: An NMR Study"

_pharmaceuticals, 2021, doi:10.3390/ph14080732_

Round 1
Reviewer 1 Report
This paper reports the study of monomeric units of the amyloid-beta (1-42) peptide in the presence of different HFIP/water concentrations. Results have been obtained using a NMR approach and then employed to perform classical MD simulations. Conclusions are supported by results that are adequately explained.
I have some minor concerns:
in figure 2 there is a cluster of points at psi = 70 and phi = -90 between the helical and sheet region. The points in this rage are much clusterized respect to other points. Why? Do they belong to a particular secondary structure of the amyloid?
lines 173-174 and figure 7: in the text authors says that there are sheets in the 4-14 and 32-38 regions confirmed by the DSSP analysis, but they are not present in the picture.
lines 186-190: how did author performed the simulation where the protein changes its secondary structure from helix to beta-sheet? In the methods it is reported only one kind of simulation, but several previous studies employed metadynamics (such as https://doi.org/10.1038/srep15449) or other enhanced sampling techniques to see the transition. If authors see the transition without any bias, this should mean that the configuration used as input is not in a stable state.
lines 113-114: the sequence KLVFFA is one that is more studied as model for beta-sheets also for amyloid. In the text authors says that these region from V18 to V24 has a net tendency to be in a helix conformation.
Did the authors tried to see the conformational changes of amyloid-beta with HFIP when the fibrillation process starts?
Author Response
We thank the reviewer for the comments. The answer can be found in the attached word file.

Reviewer 2 Report
The authors investigated the conformation of Aβ(1-42) peptide in 50/50 hexafluoroisopropanol/water v/v solvent. The amyloid peptide is arranged in regular turn-helix-turn segments, with the C-terminal being highly flexible and dynamic. Molecular dynamics simulations in water showed that this conformation quickly evolves in the regular β-strand structure characteristic of fibril aggregates.
This is an interesting study providing important insights into current knowledge on pathophysiology of Alzheimer’s disease, which is one of the most common neurodegenerative diseases. The manuscript is well written, and I do not have any critical comments.
Minor issues and suggestions to strengthen this manuscript are raised as follows:
- The authors should use abbreviations carefully throughout the manuscript. For example, “HFIP” and “NMR” in the abstract should be spelled out at their first appearance in the abstract
- An issue regarding the commonalities of pathophysiology with other amyloidosis will attract broader range of researchers. For example, proteolytic cleavage is also an important process of transthyretin amyloid fibril formation (Neurol Ther 2020; 9: 317-333). This issue should be incorporated in the introduction section, by citing this article.
Author Response

(The authors gave the same response as above.)

Reviewer 3 Report
The manuscript entitled “Exploring the early stages of the amyloid Aβ(1-42) peptide aggregation process: an NMR study” by Santoro et al. reported an NMR conformational analysis of Aβ(1-42) in 50/50 HFIP/water v/v solvent. The NMR results have been, then, confirmed by MD simulations.
This is an interesting study, because of the fine accordance between experimental and computational data allowed to consider with confidence the structural hints of amyloid Aβ(1-42) peptide.
The paper is well written and experimental details appear to be well described.
However, I have a concern relative to the aggregation state of the peptide. I ask the authors how they evaluated the aggregation state of the peptide and why they did not acquire DOSY spectra for the evaluation of the aggregation state.
Author Response

(The authors gave the same response as above.)
